# RAO DIFFERENTIAL PRIVACY

## ABSTRACT

Differential privacy (DP) has recently emerged as a definition of privacy to re-
lease private estimates. DP calibrates noise to be on the order of an individual's
contribution. Due to this calibration, a private estimate obscures any individual
while preserving the utility of the estimate. Since the original definition, many
alternate definitions have been proposed. These alternates have been proposed for
various reasons including improvements on composition results, relaxations, and
formalizations. Nevertheless, thus far nearly all definitions of privacy have used
a divergence of densities as the basis of the definition. In this paper we take an
information geometry perspective towards differential privacy. Specifically, rather
than define privacy via a divergence, we define privacy via the Rao distance. We
show that our proposed definition of privacy shares the interpretation of previous
definitions of privacy while improving on sequential composition.

## 1 INTRODUCTION AND MOTIVATION

In the past few decades data has become increasingly accessible. This accessibility has come hand
in hand with concerns on the privacy practices in place on the security of ones data. Differential
Privacy (DP, Dwork et al., 2006b) is a statistical definition of privacy that has become a de facto
framework for protection of data. DP is compelling, among other reasons, as the amount of noise
injected into the summary of interest is on the order of an individuals contribution. This property of
DP lends itself to the interpretation of the concealment of any particular individual while preserving
the utility of the sample statistic.

DP is not a particular procedure but rather an attribute of a random mechanism. A mechanism is,
generally, a probability density or algorithm where the scale parameter is chosen in such a way to
satisfy the definition of DP. Loosely speaking, DP states that the mechanism is not too different from
itself under adjacent datasets where adjacent datasets are datasets of the same size but with exactly
one observation replaced by an arbitrary observation. Due to its simplicity, interpretability, and
practicality, DP has been utilized by the U.S. Census bureau (Abowd, 2018) and is broadly useful
for government agencies (Drechsler, 2023).

A core component of DP is the "privacy budget" which, roughly speaking, both controls the amount
of noise and determines the amount of privacy loss in the analysis. Keeping track of the budget in an
analysis is crucial for maintaining privacy. Two important properties of DP are (sequential) compo-
sition and post-processing. The former refers to the composition of budgets when performing more
than one analysis on the same dataset. That is, often a practitioner requires more than one summary
from the data and hence each respective privacy budget needs to be combined in some manner, for
instance by addition. The total budget is that of interest and is computed by a rule of composition.
The latter, post-processing, simply stated, states a practitioner can alter a private estimate arbitrarily
without revealing any additional information of the underlying data. Post-processing is crucial as
protection of data is the goal and hence post-processing must be allowed.

Since the initial definition of DP, many alternative definitions and relaxations have been proposed.
These alternatives either sought to accommodate previously unaccommodated mechanisms or to
improve on the composition rules for budgets. For instance, pure DP (Dwork et al., 2006b) cannot
accommodate the Gaussian mechanism but approximate DP (Dwork et al., 2006a), a relaxation of
pure DP, does accommodate it. Zero-concentrated DP (zCDP, Bun & Steinke, 2016) is a relaxation
of DP which states that the mechanism under adjacent datasets are similar under $\alpha$ Rényi divergence
for all $\alpha$. Rather than require the $\alpha$ Rényi divergence to be bounded by all $\alpha$, Mironov (2017) used $\alpha$

as a parameter itself to generalize zCDP to Rényi differential privacy. Lastly, we mention Kullback-Leibler DP (Barber & Duchi, 2014; Cuff & Yu, 2016) which measures similarity of the mechanism via the Kullback-Leibler divergence. These definitions are all based on divergences, although the connection between the first definitions and divergences was not discovered until later on. There are many more definitions of privacy and we refer to Desfontaines & Pejó (2019) for a review.

In essence, most, if not all, previous definitions of privacy share the same idea: *the random mechanism is not too different from itself over all possible adjacent datasets*. The main difference in these definitions is how they measure said similarity. Due to this interpretation it is to be expected that the previous literature has been drawn to divergences of densities as a measure of similarity. In statistics, data science, information theory, and machine learning the most commonplace divergence is likely the Kullback-Leibler (KL) divergence. Further, the Rényi divergence is indeed a generalization of the KL divergence. These are surely attractive but the KL and Rényi divergence measure the difference *from* a density rather than *between* densities. Arguably, a natural way to measure dissimilarity is with a proper distance metric. No previous definitions of DP, however, have considered the use of a proper metric of densities. We, thus, introduce the first definition of differential privacy defined in terms of a distance metric for densities.

The main contributions of this paper are thus:

1. We introduce Rao differential privacy, the first definition of differential privacy defined in terms of a distance metric. Namely, we utilize the Rao distance metric of densities as our measure of similarity.

2. We show that our definition respects post-processing and determine the rule for composition, two crucial components of differential privacy.

3. We determine the privacy parameters for the Gaussian and Laplace mechanisms, the two most prolific privacy mechanisms, under our framework.

4. We determine the privacy parameters of the Generalized Gaussian mechanism, a recent, attractive mechanism in privacy.

## 2 BACKGROUND

In this section we introduce all the necessary background and notation for our methodology. We first survey the landscape of the state of the art, divergence based differential privacy definitions. We do not include all definitions for brevity, however, we include those most relevant to the manner at hand and point to Desfontaines & Pejó (2019) for a thorough review. Next, we introduce differential geometry and Riemannian manifolds as this is foundational to the distance of densities in our definition. Lastly, we motivate and define the Rao distance for parametric probability densities (Rao et al., 1945). The first subsection is, admittedly, disjoint from the latter subsections as the goal of this paper is to bridge these ideas. For more details on Riemannian geometry we refer to Do Carmo (1992) and Lee (2018).

### 2.1 DIFFERENTIAL PRIVACY

In this subsection we focus on definitions of differential privacy (DP). We mention some specific mechanisms and properties of DP here but formalize them in subsequent sections. This delay in formalisms is to allow a direct comparison to our proposed counterparts.

A key component to defining DP is the idea of adjacent datasets. Let $D = \{x_1, x_2, \ldots, x_n\}$ denote a dataset of size $n$ such that $x_i \in \mathcal{X}$. Two datasets $D, D'$ are said to be *adjacent* if they differ in exactly one observation and $|D| = |D'|$; this is denoted as $D \sim D'$. Let $f_D$ denote a random mechanism, where the subscript $D$ represents the dependence on the dataset. The original definition of differential privacy is as follows.

**Definition 2.1** (Dwork et al. (2006b)). A random mechanism $f_D$ is said to be $\epsilon$-differentially private if for all $D \sim D'$

$$\log\left(\frac{\int_S d\mu f_D}{\int_S d\mu f_{D'}}\right) \leq \epsilon$$

where $\epsilon$ is a pre-specified parameter referred as the privacy budget, $S$ is any measurable set, and $\mu$ is the base measure.

The authors referred to this as the densities being $\epsilon$-indistinguishable under adjacent datasets. It is now more commonly stated that the mechanism satisfies *pure differential privacy*, pure-DP. Roughly speaking, a mechanism is a density where the order of the spread parameter, $\sigma$, is tailored to satisfy the inequality of Definition 2.1. The inequality in Definition 2.1 is more commonly, and equivalently, expressed as

$$\int_S d\mu f_D \le \exp(\epsilon) \int_S d\mu f_{D'}.$$

The former formulation lends itself to a direct interpretation on bounding the log-likelihood over adjacent datasets. The latter, while equivalent, lends itself to the interpretation that the densities are multiplicatively bounded by some constant over adjacent datasets. These interpretations are at the core of DP research and roughly state that the random privacy mechanism is "not too different" from itself under adjacent datasets. This is more clear when we notice that for small $\epsilon$, $\exp(\epsilon) \approx 1 + \epsilon$.

This definition of privacy is quite attractive, for reasons we elaborate on shortly, but an immediate shortcoming is that the Gaussian density cannot be bounded by any $\epsilon$. To achieve an accommodation of the Gaussian density, the definition of privacy must be relaxed. Fortunately, the latter inequality is well suited for a generalization and relaxation. Dwork et al. (2006a) introduced *approximate DP* which is defined as follows.

**Definition 2.2** (Dwork et al. (2006a)). A mechanism is said to be $(\epsilon, \delta)$-DP if for all adjacent datasets $D \sim D'$

$$\int_S d\mu f_D \le \exp(\epsilon) \int_S d\mu f_{D'} + \delta$$

where $\epsilon > 0$ and $\delta \in (0, 1)$ are pre-specified parameters.

An interpretation $\delta$ is that pure-DP is held with probability $1 - \delta$; as Mironov (2017) and Mironov et al. (2009) note, however, this interpretation can be difficult to formalize.

The attraction to DP can partially be attributed to the following three properties: transparency, composition, and post-processing. *Transparency* refers to the fact that the data curator can reveal the private estimate(s), the privacy mechanism, and privacy budget without revealing the non-private estimate. (Sequential) *composition*[1] refers to the combination of privacy budgets from multiple analysis conducted on the same dataset. For instance, if a data curator wishes to release more than one private statistical summary from a dataset, each analysis has its own privacy budget and composition determines the total privacy budget incurred. Lastly, *post-processing* refers to the ability to apply any deterministic or random function to a private estimate without being able to gain any additional information on the original dataset. These properties are fundamental to privacy which the former two definitions uphold and hence any subsequent definitions must satisfy these properties as well.

We next mention definitions of DP directly based on divergences. One requires the definitions of Rényi and Kullback-Leibler divergence which we include in C for completeness. With these divergences in mind, we introduce Kullback-Leibler differential privacy. The definition is as follows.

**Definition 2.3** (Barber & Duchi (2014); Cuff & Yu (2016)). A random mechanism $f_D$ said to satisfy Kullback-Leibler DP if

$$D_{KL}(f_D \| f_{D'}) \le \gamma$$

for all $D \sim D'$ and a pre-specified $\gamma$.

This definition of privacy is one of the first which directly defined privacy in terms of a divergence of densities. Interestingly, while the authors compared pure-DP and approximate-DP to KL-DP, they appear to not have made any claims on them being a divergence themselves. More on this shortly. Zero concentrated DP (zCDP) (Bun & Steinke, 2016) is the next definition we mention. This definition is based on the Rényi divergence.

---

[1]We only consider sequential composition and hence refer to it simply as composition. However we note there are other forms of composition such as parallel composition.

**Definition 2.4** (Bun & Steinke (2016)). A randomized mechanism $f_D$ is said to satisfy $(\xi, \rho)-$zero concentrated differentially privacy if for all $D \sim D'$ and all $\alpha \in (1, \infty)$,

$$D_\alpha(f_D \| f_{D'}) \leq \xi + \rho\alpha$$

where $D_\alpha(\cdot \| \cdot)$ is the $\alpha$-Rényi divergence. For $(0, \rho)$ a mechanism satisfies $\rho$-zCDP.

A limitations with $\rho$-zCDP the mechanism is required to by bounded by the Rényi divergence for all $\alpha$ (the parameter in the definition of Rényi divergence). This requirement is quite strict as larger $\alpha$ translate to stricter bounds on rare events.

Moving from the KL divergence to the Rényi divergence is a natural generalization however requiring the inequality to hold for all $\alpha$ is quite strong. Thus, Mironov (2017) relaxed the aforementioned limitation by having $\alpha$ be a parameter of the definition rather than a restriction. This lead to Rényi differential privacy defined as follows.

**Definition 2.5** (Mironov (2017)). A randomized mechanism $f_D$ is said to satisfy $\zeta-$Rényi differential privacy of order $\alpha$

$$D_\alpha(f_D \| f_{D'}) \leq \zeta$$

for all $D \sim D'$ and pre-specified $\zeta$.

Having these few definitions we can see that these are intricately connected. The connection between zCDP and Rényi DP is clear as they both use the same divergence. Also, as $\alpha \to 1$ Rényi DP is Kullback-Leibler DP. Further Mironov (2017) noted that pure DP can be expressed as the Rényi DP for $\alpha \to \infty$. Lastly, approximate-DP has also been shown to be tied to the max-Kullback-Leibler stability. That is, all the presented definitions, including the original definitions, are in fact divergence based.

Desfontaines & Pejó (2019) have a thorough review on the vast landscape of definitions of privacy as well as the connection between said definitions. While this is only a few of the many definitions of differential privacy, we note that all presented definitions require the mechanism to not be too different from itself over adjacent datasets. This similarity is quantified by the use a statistical divergence. While these definitions are motivated and constructed in vastly different manners, they are deeply connected. To our knowledge, however, there seems to be no definitions of privacy which utilize a proper distance metric of densities for quantification of similarity of mechanism. That is our goal as divergences measure a distance *from* a density rather than between densities. Following we introduce differential geometry as it is a fundamental in the construction of the distance we intend to use.

## 2.2 RIEMANNIAN MANIFOLDS

Let the tuple $(\mathcal{M}, g)$ denote a complete Riemannian manifold. Here $\mathcal{M}$ is a smooth manifold and $g$ is the Riemannian metric tensor. At each point $p \in \mathcal{M}$ we have a tangent space $T_p\mathcal{M}$. The tangent space is the span of all vectors which are tangent to the manifold, i.e. $T_p\mathcal{M} = \text{span}\{\dot{\alpha}(0) | \alpha : (-\epsilon, \epsilon) \to \mathcal{M}, \alpha(0) = p\}$ with $\alpha$ being a smooth path. The Riemannian metric $g$ defines an inner product on each tangent space and varies smoothly on the manifold. So, $g : T_p\mathcal{M} \times T_p\mathcal{M} \to \mathbb{R}^{\geq 0}$ and is denoted as $g(u, v) = \langle u, v \rangle_{g(p)}$, or $\langle u, v \rangle_p$ for brevity, where $u, v \in T_p\mathcal{M}$ are tangent vectors. The metric $g$ expressed as a matrix has elements $g_{ij} = \langle \partial x^i, \partial x^j \rangle$ where $\partial x_1, \partial x_2, \ldots, \partial x_d$ is a set of local basis on $T_p\mathcal{M}$. The metric is invariant to choice of basis. Under this matrix representation we, further, have that $g(u, v) = \sum_{ij} g_{ij}(p) u_i v_j$ with $p$ being the footpoint on the manifold. The metric is symmetric positive definite, and hence invertible. The inverse is denoted as $g^{ij}$.

The inner product lends itself to geometric interpretations such as angles and lengths. Let $\alpha(t) : [0, 1] \to \mathcal{M}$ be a smooth path such that $\alpha(0) = p$, $\alpha(1) = q$. The length of the path, $\mathcal{L}(\alpha)$, is the cumulative instantaneous rate of change: $\mathcal{L}(\alpha) = \int_0^1 \langle \frac{d}{dt}\alpha(t), \frac{d}{dt}\alpha(t) \rangle_{\alpha(t)}^{1/2} dt$. The integrand is typically referred to as the *line element* and denoted $ds$. This denotation conveys the length of infinitesimally small vector $ds^2 = \sum_{ij} g_{ij} v_i v_j$ where $v_i$ are the local coordinates of vector $v = \dot{\alpha}$. So, we have $\mathcal{L}(\alpha) = \int_0^1 ds\, dt$ with $ds$ implicitly being a function of $t$. The distance between two points $p, q$ is the length of the smooth path joining the two points of minimal length. That is, $d(p, q) := \inf_{\alpha, \alpha(0) = p, \alpha(1) = q} \mathcal{L}(\alpha)$ with $\alpha$ being a smooth path. For the path to be of minimal

length it must be a solution to the Euler-Lagrange equations. This requirement can be difficult to accommodate, however when the dimension of the manifold is $d = 1$, there is a unique geodesic and hence no difficulty. As we will work on a single dimensional manifold, there will be no such complications.

### 2.3 RAO'S DISTANCE

With the background of Riemannian manifolds in mind, we introduce Rao's distance for parametric densities. We motivate the distance as in (Rao et al., 1945) and a more thorough assessment can be found in (Atkinson & Mitchell, 1981; Miyamoto et al., 2024).

Suppose we have a family of parametric densities defined over a parameter space $\Theta$ where $\theta \in \Theta \subset \mathbb{R}^n$. Each density $p_\theta \in \mathcal{P}_\theta$ is identified by a $\theta \in \Theta$. That is, $\mathcal{P}_\theta := \{p_\theta = p(x; \theta); \theta \in \Theta\}$ is a family of densities with $\int d\mu\, p(x) = 1$. The random variable $x$ can be either discrete or continuous, the measure in the following will simply need to be adopted to each situation. So, $\mu$ denotes the typical Lebesgue measure for continuous distributions and count measure for discrete distributions. The densities must satisfy regularity conditions which are outlined in Appendix B.

Given any two densities $p_1, p_2 \in \mathcal{P}_\theta$, it is clear that $p_1 + p_2 \notin \mathcal{P}_\theta$. Since addition does not hold, $\mathcal{P}_\theta$ is not a linear space and thus the geometry of it must be considered for a proper consideration of distance. Each density is identified by their parameters and thus the distance between the densities is the distance between the parameters in $\Theta$. That is, the Rao distance $d_R(p_1, p_2) := d_R(\theta_1, \theta_2)$ with $p_1, p_2 \in \mathcal{P}_\theta$ and $\theta_i$ being the parameters of $p_i$.

To incorporate the geometry of $\mathcal{P}_\theta$ into the computation of a distance, Rao identified the manifold structure of $\mathcal{P}_\theta$ (Rao et al., 1945). We thus require a metric $g$. First, recall the information matrix (Fisher, 1922) of a density is defined as the variance of the score as,

$$E\left(\frac{\partial \ln p(x; \theta)}{\partial \theta_i} \frac{\partial \ln p(x; \theta)}{\partial \theta_j} | \theta\right).$$

It is clear that

$$\frac{\partial \ln p(x; \theta)}{\partial \theta_i} = \frac{1}{p(x; \theta)} \frac{\partial p(x; \theta)}{\partial \theta_i},$$

and it is important to note the significance of this relationship. While the LHS is the typical representation, note the RHS clearly displays that the derivative of the log is the *relative change* of a density with respect to a parameter.

Rao noticed that the information matrix, $g_{ij}(\theta) = E\left(\frac{\partial \ln p(x; \theta)}{\partial \theta_i} \frac{\partial \ln p(x; \theta)}{\partial \theta_j} | \theta\right)$, can be designated as the metric $g$ for $\mathcal{P}_\theta$. Briefly, the information matrix is a symmetric positive definite matrix at every density and this is a necessary property of a metric. Combining this with the ideas of Section 2.2 the distance between two densities, represented by, with slight abuse of notation, $\theta_1, \theta_2$ is then given by $\int_{\theta_1}^{\theta_2} ds\, dt$. The squared-line element thus being

$$ds^2 = \sum g_{ij} \frac{d\theta_i}{dt} \frac{d\theta_j}{dt},$$

with $g$ being the information matrix.

Generally speaking, the distance between two densities can be challenging to compute. One needs to find the geodesic by satisfying the Euler-Lagrange equations, so there is a bottleneck in this computation. For instance, for general multivariate Gaussian there is no closed-form equation for the distance (Nielsen, 2023; Pinele et al., 2020). It turns out that for one parameter densities, however, most of these complications do not occur. Below we give an example on how to compute such a distance.

**Example 1.** *(Rao et al., 1945; Burbea & Rao, 1982) Consider the univariate Gaussian density* $f(x) = \frac{1}{\sqrt{2\pi\sigma^2}} \exp(-(x-\mu)^2/(2\sigma^2))$. *Here* $\theta = (\mu, \sigma)$ *is the set of parameters. The entries of the information matrix are* $g_{11} = \mathbb{E}[\frac{\partial}{\partial \mu} \log f \frac{\partial}{\partial \mu} \log f] = \mathbb{E}[(x-\mu)^2/\sigma^4] = 1/\sigma^4 \mathbb{E}[(\mu-x)^2] = 1/\sigma^2$, $g_{12} = g_{21} = \mathbb{E}[\frac{\partial}{\partial \mu} \log f \frac{\partial}{\partial \sigma} \log f] = 0$, *and* $g_{22} = \mathbb{E}[\frac{\partial}{\partial \sigma} \log f \frac{\partial}{\partial \sigma} \log f] = \mathbb{E}[-1/\sigma^2 + (x-\mu)^4/\sigma^6] = 2/\sigma^2$. *We have the information matrix, and hence the Riemannian metric is*

$$g = \begin{bmatrix} \frac{1}{\sigma^2} & 0 \\ 0 & \frac{2}{\sigma^2} \end{bmatrix}.$$

*The coordinate pair in the parameter space is $(\mu, \sigma)$, and the entries of the metric tensor give us information on how these parameters effect the distance. For instance, we note that $g_{11}$ says that, for two Gaussian densities with the same $\sigma$, they are increasingly similar for larger $\sigma$. This matches intuition as the distributions become infinitely flat as $\sigma \to \infty$, so they locally become more and more similar. Computing the distance between two normal densities is not immediately straightforward as we need to integrate*

$$\int_{(\mu_1,\sigma_1)}^{(\mu_2,\sigma_2)} \sqrt{\sum g_{ij} \partial\mu \partial\sigma} dt = \int_{(\mu_1,\sigma_1)}^{(\mu_2,\sigma_2)} \sum \sqrt{\frac{1}{\sigma^2}(\partial\mu)^2 + \frac{2}{\sigma^2}(\partial\sigma)^2} dt$$

*We can identify this line element with that of the upper Poincare half-plane so we have that*

$$d_R((\mu_1,\sigma_1),(\mu_2,\sigma_2)) = 2\sqrt{2}\log \frac{\sqrt{(\mu_1-\mu_2)^2 + 2(\sigma_1+\sigma_2)^2} + \sqrt{(\mu_1-\mu_2)^2 + 2(\sigma_1-\sigma_2)^2}}{\sqrt{(\mu_1-\mu_2)^2 + 2(\sigma_1+\sigma_2)^2} - \sqrt{(\mu_1-\mu_2)^2 + 2(\sigma_1-\sigma_2)^2}}.$$

*We note this can be algebraically rewritten. The two special cases, as noted by Rao are $d_R((\mu_1,\sigma),(\mu_2,\sigma)) = \frac{|\mu_1-\mu_2|}{\sigma}$ and $d_R((\mu,\sigma_1),(\mu,\sigma_2)) = \sqrt{2}|\log(\sigma_2/\sigma_1)|$.*

While this is clearly a complex distance, for densities with only one parameter the distance is much simpler as the geodesics are immediate. The Gaussian, for instance, can be thought of as having one parameter by setting the either the mean or variance as fixed.

# 3 RAO DIFFERENTIAL PRIVACY

In Section 2.1 we introduced some definitions of Differential Privacy. Pure, approximate, zero concentrated, and Rényi DP all share the identifying feature that they define privacy in terms of a divergence of mechanisms over adjacent datasets. In Section 2.3 we introduced the Rao distance of densities and rely on those tools for the remaining of the paper. In this section we introduce *Rao DP*, our proposed definition of privacy based on the Rao distance.

**Definition 3.1.** A random mechanism $f_D$ satisfies $\theta-$Rao differential privacy if for all adjacent datasets $D \sim D'$ we have that

$$d_R(f_D, f_{D'}) \leq \theta$$

where $d_R(\cdot, \cdot)$ is the Rao distance for densities and $\theta$ is the privacy budget, a pre-specified parameter.

As opposed to the previous approaches of differential privacy, this utilizes a proper distance metric on densities. Previous methods of privacy have been defined of divergences for their attractive properties but we show that our proposed method is not only more intuitive but also is a generalization. Comparing the formulations more directly, previous formulations bound the log-likelihood ratio of the mechanism over adjacent datasets but here we rather consider how the *relative* log-likelihood changes between the adjacent datasets.

We note that the Rao distance requires the densities to have the same support, and this is a standard assumption in DP. Importantly, the privacy loss random variable is not defined if the support of the two densities is not the same. The assumptions on the densities are outlined in Appendix B but we note they are the standard regularity conditions necessary for the information matrix entries.

## 3.1 COMPOSITION

A key property of DP is composition. Composition refers to the composition of privacy budgets when a data curator conducts more than one private analysis on a dataset. That is, given two mechanisms $f_1, f_2$ with respective privacy budgets $\theta_1, \theta_2$, composition is an assurance that combined results also satisfy the definition of privacy and how the budgets should be combined. A large motivation for introduction of novel definitions of DP is tighter bounds on composition of budgets.

To establish composition results for Rao privacy we need to consider the Rao distance between product densities. So, in Appendix A we mention standard results of distances on the product space of Riemannian manifolds; as a specific case for the Rao distance see Burbea & Rao (1982). We have that Rao distance between product densities is the square root of the sum of square Rao distances between the marginal densities, i.e.,

$$d_R(f_{1,D} \times f_{2,D}, f_{1,D'} \times f_{2,D'}) = \sqrt{d_R(f_{1,D}, f_{1,D'})^2 + d_R(f_{2,D}, f_{2,D'})^2}.$$

Table 1: Privacy definitions and their sequential composition result. Index $i$ refers to $i$th budget.

| | Budget | Composition |
|---|---|---|
| Pure DP (Dwork et al., 2006b) | $\epsilon$ | $\sum \epsilon_i$ |
| Approx DP (Dwork et al., 2006a) | $(\epsilon, \delta)$ | $(\sum \epsilon_i, \sum \delta_i)$ |
| KL DP (Barber & Duchi, 2014; Cuff & Yu, 2016) | $\alpha$ | $\sum \alpha_i$ |
| mCDP (Dwork & Rothblum, 2016) | $(\mu, \tau)$ | $(\sum \mu_i, \sqrt{\sum \tau_i^2})$ |
| zCDP (Bun & Steinke, 2016) | $(\xi = 0, \rho)$ | $(0, \sum \rho_i)$ |
| Rényi DP (Mironov, 2017) | $(\alpha, \epsilon)$ | $(\alpha, \sum \epsilon_i)$ |
| GDP (Dong et al., 2022) | $\mu$ | $\sqrt{\sum \mu_i^2}$ |
| **Rao DP (Ours)** | $\theta$ | $\sqrt{\sum \theta_i^2}$ |

**Lemma 3.2** (Composition). *Given the above, two mechanisms $f_{1,D}$ and $f_{2,D}$ with privacy budgets $\theta_1$ and $\theta_2$, respectively, the total privacy budget is $\sqrt{\theta_1^2 + \theta_2^2}$.*

Compared to pure DP, the composition here is much tighter. Under pure DP the composition of the budgets $\epsilon_1$ and $\epsilon_2$ is simply the sum $\epsilon_1 + \epsilon_2$. Clearly we have that $\epsilon_1 + \epsilon_2 > \sqrt{\epsilon_1^2 + \epsilon_2^2}$, but this is only useful when the budgets mean the same thing in different definitions of privacy. So, if a mechanism satisfies both pure DP and Rao DP and one needs to conduct multiple analyses, then utilizing Rao-DP one requires less total budget for comparable statistical utility. Table 1 lists definitions of privacy and their respective rule for composition. We did not introduce GDP (Dong et al., 2022) in 2.1 as it is not divergence based, but it has the same composition as Rao DP. We elaborate on GDP in C.1.

### 3.2 POST-PROCESSING

Another important property of differential privacy is post-processing. We first summarize the previous notions of post-processing to draw a comparison. Let $f(x; D)$ be a random mechanism that satisfies $(\epsilon, \delta)$-DP and $m$ be a random map; we have that $m(f(x; D))$ also satisfies $(\epsilon, \delta)$-DP (Dwork et al., 2006b).

To show that Rao DP is immune to post-processing, we appeal to the following. Given two densities $p_1, p_2$ the Rao distance, using the square root transformation, $p(x) \mapsto \sqrt{p(x)}$, the Rao distance is then $d_R(p_1, p_2) = 2 \cos^{-1} \langle \sqrt{p_1}, \sqrt{p_2} \rangle$ where $\langle \cdot, \cdot \rangle$ is the $\mathbb{L}^2$ inner product (Amari & Nagaoka, 2000; Ay et al., 2017). Noting that $\int dx \left( \sqrt{p} \right)^2 = 1$ we see this transformation embeds the densities onto the positive orthant of a sphere.

**Theorem 3.3.** *Let $f_D$ be a random mechanism that satisfies Rao DP for some $\theta$ and $\varphi$ be an arbitrary random function. We have that $\varphi \circ f_D$ also satisfies Rao DP under the same $\theta$.*

The proof is in Appendix D but follows directly from the square root transformation and the $\mathbb{L}^2$ inner product properties. In the next sections we determine the rate parameters for common mechanisms, Laplace and Gaussian, and the Generalized Gaussian under Rao DP.

## 4 MECHANISMS

In the previous sections, we have built the necessary tools to define Rao-DP. We showed Rao-DP satisfies composition and is immune to post-processing. Lastly, here we determine the privacy parameters of common mechanisms. Thus, in this section we show that the Laplace, Gaussian, and the Generalized Gaussian in Appendix E satisfy Rao-DP. We utilize the closed form Rao distances for densities from Miyamoto et al. (2024) and include derivations when necessary.

**Definition 4.1.** The $\eta$-sensitivity of a function $h : D \to \mathcal{S}$ is, $\Delta_\eta(h) = \sup_{D \sim D'} d_\eta(h(D), h(D'))$ with $d_\eta$ being a distance function.

This is typically referred to as *global sensitivity* but we simply refer to this as the sensitivity as we do not consider other notions of sensitivity such as local sensitivity. The particular distance for the sensitivity is dependent on both the mechanism and the space $\mathcal{S}$ and the most common distances are the $\mathbb{L}^1$ and $\mathbb{L}^2$ distances. Further, $\mathcal{S}$ is typically $\mathbb{R}^m$ but with recent advances on DP for Riemannian

manifold, the distance is of that space (Reimherr et al., 2021). While its necessary to be cognizant of $\eta$, the distance is typically clear from the context, so we drop the superfluous notation.

### 4.1 LAPLACE MECHANISM

The Laplace mechanism was the first mechanism introduced in (Dwork et al., 2006b). The Laplace density, also known as the double exponential, is defined as $f(x; \mu, b) = (2\sigma)^{-1} \exp\{|x - \mu|/\sigma\}$ where $\mu$ is the expected value and $\sigma$ is the spread parameter. Let us determine the Rao distance between Laplace densities with fixed, equal $\sigma$ as this is necessary to show the mechanism satisfies Rao DP.

**Theorem 4.2.** *Let $f_1(x; \mu_1, \sigma)$ and $f_2(x; \mu_2, \sigma)$ be two Laplace densities with the same scale parameter. The Rao distance between $f_1$ and $f_2$ is,*

$$d_R(f_1(x; \mu_1, \sigma), f_2(x; \mu_2, \sigma)) = \sqrt{\sum_i \frac{|\mu_{i,1} - \mu_{i,2}|^2}{\sigma}}.$$

*Proof.* The proof is nearly identical to Example 1 with the information matrix being $g_{11} = \frac{1}{\sigma^2}$. $\square$

**Definition 4.3.** The Laplace mechanism to release a private version of summary $h(D)$ takes the form $f(x; D) = (2\sigma)^{-1} \exp\{|x - h(D)|/\sigma\}$.

In the case of Euclidean data, $D \subset \mathbb{R}$, one can equivalent see that the Laplace mechanism "adds" noise as $h(D) + f(x; \mu = 0, \sigma)$. While this is a convenient representation, we avoid this formulation as addition is not defined on non-linear spaces such as Riemannian manifolds.

**Theorem 4.4.** *The Laplace mechanism with $\sigma \geq \Delta/\theta$ satisfies $\theta$-Rao Privacy.*

*Proof.* This follows semi-directly from Theorem 4.2. We have that $d_R(f_D, f_{D'}) = \frac{|h(D) - h(D')|}{\sigma}$. We thus need $\frac{|h(D) - h(D')|}{\sigma} \leq \theta$ for all $D \sim D'$. So,

$$\frac{|h(D) - h(D')|}{\sigma} \leq \frac{\Delta}{\sigma}$$
$$\implies \sigma \geq \Delta/\theta.$$

$\square$

Under the pure DP framework the spread parameter must be $\sigma \geq \Delta/\epsilon$. Thus we have the following result.

**Corollary 4.5.** *The Laplace mechanism that satisfies $\epsilon$ pure DP satisfies Rao DP with $\theta = \epsilon$.*

It can be thus argued this is just a reparameterization of privacy since for a single query $\theta = \epsilon$ the mechanism is exactly the same. However, the difference in the framework lies when there are multiple queries and hence composition is necessary. That is given two queries sanitized with the Laplace in pure DP with budgets $\epsilon_1, \epsilon_2$ their total budget is $\epsilon_1 + \epsilon_2$ which is greater than $\sqrt{\epsilon_1^2 + \epsilon_2^2}$, the total privacy budget under Rao DP. Thus, for the Laplace mechanism under Rao DP has tighter composition than pure DP with the same properties such a transparency and immunity to post-processing.

### 4.2 GAUSSIAN MECHANISM

The Gaussian mechanism was introduced in Dwork et al. (2006a). Pure DP cannot accommodate the Gaussian mechanism for any $\epsilon$, so pure DP was relaxed to approximate DP. The Gaussian, normal, density is defined as $f(x; \mu, \sigma) = (2\pi\sigma^2)^{-1/2} \exp\{(x - \mu)^2/2\sigma^2\}$ where $\mu$ is the expected value and $\sigma$ is the spread parameter and standard deviation. Again, we first determine the Rao distance between Gauss densities with fixed, equal $\sigma$ as this is necessary to show the mechanism satisfies Rao DP.

**Theorem 4.6.** *Let $f_1(x; \mu_1, \sigma)$ and $f_2(x; \mu_2, \sigma)$ be two Gaussian densities with the same scale parameter. The Rao distance between $f_1$ and $f_2$ is,*

$$d_R(f_1(x; \mu_1, \sigma), f_2(x; \mu_2, \sigma)) = \frac{|\mu_1 - \mu_2|}{\sigma}.$$

We refer to Example 1 for a proof. We are now ready to define the Gaussian mechanism.

**Definition 4.7.** The Gaussian mechanism to release a private version of summary $h(D)$ takes the form $f(x; D) = (2\pi\sigma^2)^{-1/2} \exp\{(x - h(D))^2/2\sigma^2\}$.

**Theorem 4.8.** *The Gaussian mechanism with $\sigma \geq \Delta/\theta$ satisfies $\theta$-Rao DP.*

The proof is identical to the case for the Laplace mechanism. Again we compare our definition to the state of the art definitions.

**Corollary 4.9.** *The Gaussian mechanism that satisfies $\mu$-GDP satisfies Rao DP with $\theta = \mu$. The Gaussian mechanism that satisfies $(\epsilon, \delta)$-DP satisfies Rao DP with $\theta = \epsilon/\sqrt{2\log(1.25/\delta)}$.*

First, for the Gaussian mechanism, GDP and Rao DP act exactly the same. This is quite remarkable as the formulation of GDP is entirely different than that of our proposed definition and they do not rely on a divergence nor distance at all. Further, we have the same composition result. We make some remarks on this in Section C.1. As opposed to approximate DP, we do not have a second parameter such as $\delta$. This is a strength of our framework since, as noted previously, this $\delta$ is the probability of privacy leakage. Our formulation does not allow such a case. Further, it seems there are many combinations of $(\epsilon, \delta)$ which give the same $\theta$. This is the same situation that arises in GDP. Again though the composition of approximate DP is simple addition while our composition is, in a sense, "tight."

## 5 DISCUSSION

We have introduce a novel definition of differential privacy, Rao DP. As opposed to previous definitions of DP which are defined in terms of divergences we utilize a distance metric namely the Rao distance of densities. This reliance on a distance manifests itself in a tight composition of budgets. We show that the Laplace and Gaussian mechanism both satisfy Rao DP, these being cornerstone mechanisms of DP. Lastly we mention the generalized Gaussian mechanism, a generalization of the Laplace and Gaussian mechanisms, in Appendix E.

It is not entirely surprising that using Rao's distance we have a "tight" composition. Composition effectively measures the similarity of product densities and since divergences typically do not satisfy the triangle inequality, then the divergence of product densities incurs an inflation. Further divergences are, roughly speaking, approximately the square root of distances and as the divergences are small here this square rooting inflates the true difference. Thus, the divergence based DP definitions are surely to suffer from inflated budget composition.

GDP has the same tight composition as our proposed definition, so in C.1 we discuss a possible connection between Rao DP and GDP. A peculiarity of GDP is that for a mechanism to satisfy GDP one is required to compare to the trade off function of the Gaussian. While the importance of the Gaussian cannot be understated, it has an arbitrary nature to it.

There are many natural extensions which should be considered. Namely we only considered continuous densities, but one can also consider discrete distributions. Further, we focused on real valued densities but this definition can be extended to non-Euclidean spaces. Lastly, we only showed results for the most foundational mechanisms but there are many other mechanisms which one can consider.

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

## A    MANIFOLD NOTES

Again, we note that many of these results are standard in differential geometry and we make no claims of these being original. We refer to classic texts such as Do Carmo (1992) and Lee (2018) for more details.

For the path to be of minimal length it must be a solution to the Euler-Lagrange equations,

$$\sum_i g_{ik}(x)\frac{d^2 x_i}{dt^2} + \sum_{ij}[ij,k]\frac{dx_i}{dt}\frac{dx_j}{dt} = 0 \; k = 1, 2, \ldots, d$$

where $[ij,k]$ are the Christoffel symbols[2] defined as

$$[ij,k] = \frac{1}{2}\left(\frac{\partial g_{ik}(x)}{\partial x_j} + \frac{\partial g_{ik}(x)}{\partial x_i} - \frac{\partial g_{ij}(x)}{\partial x_k}\right) \; i,j,k = 1, 2, \ldots, d.$$

The Euler-Lagrange equations tell us that a geodesic is entirely characterized by its starting point and initial velocity.

*Product Manifold Notes for Composition* To establish composition results for Rao privacy we need to consider the Rao distance between product densities. Given two Riemannian manifolds $(\mathcal{P}_1, g_1)$ and $(\mathcal{P}_2, g_2)$, with the product density $\mathfrak{f} = f_1 \times f_2$ is an element of the *product manifold* $\mathcal{P} = \mathcal{P}_1 \times \mathcal{P}_2$. The product manifold is equipped with the Riemannian metric,

$$g = \begin{bmatrix} g_1 & 0 \\ 0 & g_2 \end{bmatrix}$$

with the 0's being matrices of zeros of suitable size and $g_1, g_2$ being the matrix representation of the metric from the respective space. On product manifolds, the distance between any two elements can be computed utilizing the Pythagorean Theorem. That is,

$$d_{\mathcal{P}}(\mathfrak{f}(0), \mathfrak{f}(1)) = \sqrt{d_{\mathcal{P}_1}(f_1(0), f_1(1))^2 + d_{\mathcal{P}_2}(f_2(0), f_2(1))^2}.$$

## B    INFORMATION MATRIX

Let $f(x; \theta)$ be a parametric probability density and $\mathcal{P}_\theta$ be the space of all such densities. The following regularity conditions are necessary.

- The partial derivatives of $f(x; \theta)$ w.r.t. $\theta$, $\frac{\partial f(x;\theta)}{\partial \theta_i}$, must exist almost everywhere for all $i$.
- Integration and differentiation can be interchanged. That is, $\int dx \frac{\partial f(x;\theta)}{\partial \theta_i} = 0$.
- The support of $f(x; \theta)$ does not depend on $\theta$.

Further, the information matrix $g$ is symmetric positive definite.

## C    DIFFERENTIAL PRIVACY DETAILS

As we define Rényi DP and Kullback-Leibler DP in 2.1, we define these divergences below.

**Definition C.1.** For two probability densities $f_1$ and $f_2$, the Rényi divergence of order $\alpha > 1$ from $f_1$ is,

$$D_\alpha(f_1 \| f_2) = \frac{1}{\alpha - 1} \log \mathbb{E}\left(\frac{f_1(x)}{f_2(x)}\right)^\alpha.$$

While this is defined for $\alpha > 1$, one can take the limit as $\alpha \to 1$ and show the limit is the Kullback-Leibler divergence.

**Definition C.2.** For two probability densities $f_1$ and $f_2$, the Kullback-Leibler divergence from $f_1$ is defined as

$$D_{KL}(f_1 \| f_2) = \int f_1(x) \log \frac{f_1(x)}{f_2(x)} dx.$$

---

[2]The Christoffel symbol is more commonly denoted as $\Gamma_{ij}^k$. We use $[ij, k]$ as it aligns with the notation in Rao et al. (1945), a foundation of the current work.

### C.1 GAUSSIAN DIFFERENTIAL PRIVACY

Here we give more details on $\mu$-GDP (Dong et al., 2022). We note that the authors mention that one of their goals was to move away from "divergence" based similarity, as is ours. Their goal, however, is to reformulate DP entirely via the lens of hypothesis testing as described next.

One way to interpret approximate DP is via hypothesis testing. One can consider the indistinguishability as a test between the hypotheses $H_0$ :the true underlying dataset is $D$ versus $H_0$ :the true underlying dataset is $D'$ at significance level $\alpha = \delta$ (Wasserman & Zhou, 2010). This idea was formalized into a definition of DP by Dong et al. (2022) in what they refer to as Gaussian differential privacy ($\mu-$GDP). This way of defining DP is slightly different than predecessors as it does not draw directly on a measure of similarity such as a divergence. First, they define their form of comparison via trade-off functions as,

**Definition C.3** (g-DP). A randomized mechanism $f(x; D)$ is said to be $g-$differentially private if for all $D \sim D'$ and all

$$T(f(x; D), f(x; D')) \geq g$$

where $T$ is a trade-off function. A function $g : [0, 1] \to [0, 1]$ is a trade-off function if and only if $g$ is convex, continuous, non-increasing, and $g(x) \leq 1 - x$ for all $x \in [0, 1]$.

One can then pick a particular density for which one needs to compute their trade-off function. The authors, as the name implies choose the Gaussian density. Their definition states,

**Definition C.4.** A mechanism $f$ is said to satisfy $\mu-$Gaussian differential privacy if for all adjacent datasets $D \sim D'$

$$T(f(x; D), f(x; D')) \geq G_\mu$$

where $G_\mu$ is the Gaussian trade off function and $\mu$ is a pre-specified privacy budget.

So, $\mu$-GDP states that it is harder to distinguish the mechanism under adjacent datasets than is to distinguish the standard normal from the normal $N(\mu, 1)$. A drawback with this definition is that it is entirely defined in relation to the normal distribution.

There is a connection between GDP and Rao DP for Gaussians. First, we note that for Gaussians with the same standard deviation have $d_R((\mu_1, \sigma), (\mu_2, \sigma)) = |\mu_1 - \mu_2|/\sigma$. GDP considers the trade-off between $N(0, 1)$ and $N(\mu, 1)$. The distance between these normals is exactly $\mu$ which is the privacy parameter of GDP; that is, $d_R((\mu_1 = 0, \sigma = 1), (\mu_2 = \mu, \sigma = 1)) = |\mu|$. Lastly, we mention that Rao suggested using the Rao distance as the numerator of a test statistic which further connects Rao DP to GDP as GDP considers the space of all hypothesis tests. This would be an interesting avenue to further investigate.

## D POST-PROCESSING PROOF

*Proof of Theorem 3.3.* We prove the theorem for arbitrary deterministic functions as a random functions can be expressed in terms of deterministic functions. Let $\varphi$ be an arbitrary deterministic function and $f(x)$ be our mechanism. By definition we have $d_R(f(x; D), f(x; D')) \leq \theta$ and thus $\int \sqrt{f(x; D)f(x; D')} \leq \theta$. Further note that,

$$\int_{\mathcal{X}} \varphi(f(x; D)) + \varphi(f(x; D'))dx = \int_{\varphi^{-1}(\mathcal{X})} f(x; D) + f(x; D')dx$$

where $\varphi^{-1}(\mathcal{X})$ is the pre-image. Thus we have that

$$\int \left( \sqrt{\varphi(f(x; D))} + \sqrt{\varphi(f(x; D'))} \right)^2 dx - 2 \int \sqrt{\varphi(f(x; D))\varphi(f(x; D'))}dx$$

$$= \int_{\varphi^{-1}(\mathcal{X})} \left( \sqrt{f(x; D)} + \sqrt{f(x; D')} \right)^2 dx - 2 \int_{\varphi^{-1}(\mathcal{X})} \sqrt{f(x; D)(f(x; D')}dx$$

As so,

$$\int \sqrt{\varphi(f(x; D))\varphi(f(x; D'))} \le \int \sqrt{f(x; D)f(x; D')} \le \theta.$$

Note that $(\sqrt{x} + \sqrt{y})^2 - 2\sqrt{xy} = x + y$. Since this holds for all $D \sim D'$ we have by symmetry that $\|\varphi(f(x; D)) - \varphi(f(x; D'))\| \le \|f(x; D) - f(x; D')\|$. We thus have the required result. $\quad\square$

Conceptualizing the space of half densities as a sphere, the Rao distance is length of the great circle connecting the two density parameters.

# E    GENERALIZED GAUSSIAN MECHANISM

Lastly we mention a mechanism which is itself a generalization of the Laplace and Gaussian mechanism. The generalized Gaussian mechanism is a recent flexibly mechanism based on the generalized Gaussian density Liu (2018). The generalized Gaussian density takes the form $f^N(x; \mu, \sigma) = (2\sigma\Gamma(1/N)/N)^{-1} \exp\{|x - \mu|^N/\sigma\}$ where $\Gamma(\cdot)$ is the gamma function.

**Theorem E.1.** *Let $f_1^N(x; \mu_1, \sigma)$ and $f_2^N(x; \mu_2, \sigma)$ be two Generalized Gaussian densities with the same scale parameter. That is $f_i(x; \mu, \sigma) = (2\sigma\Gamma(1/N)/N)^{-1} \exp\{|x - \mu_i|^N/\sigma\}$. The Rao distance between $f_1$ and $f_2$ is*

$$d_R(f_1(x; \mu_1, \sigma), f_2(x; \mu_2, \sigma)) = \frac{|\mu_1 - \mu_2|}{\sigma} \left( \frac{N\Gamma(2 - 1/N)}{\Gamma(1 + 1/N)} \right)^{1/2}$$

The derivation of the information matrix can be found in (Miyamoto et al., 2024).

**Definition E.2.** The generalized Gaussian mechanism of degree $N$ to release a private version of summary $h(D)$ takes the form $f^N(x; D) = (2\sigma\Gamma(1/N)/N)^{-1} \exp\{|x - h(D)|^N/\sigma\}$.

This mechanism has the additional parameter of $N$ which controls the spread about $h(D)$. Of course one cannot freely control the concentration parameter to get free privacy and the following theorem shows the spread parameter increases with $N$.

**Theorem E.3.** *The Generalized Gauss mechanism with $\sigma \ge \Delta \left( \frac{N\Gamma(2-1/N)}{\Gamma(1+1/N)} \right)^{1/2} /\theta$ satisfies $\theta$-Rao Privacy.*

*Proof.* We have that $g_{11} = \frac{N\Gamma(2-1/N)}{\sigma^2\Gamma(1+1N)}$. This is again, independent of $\mu$ and thus the distance is simply $|\mu_1 - \mu_2|$ multiplied by $g_{11}$. $\quad\square$

For a more thorough discussion of the importance of the Generalized Gauss mechanism, we refer to Liu (2018). Further, we note that for $N = 1$ and $N = 2$ we get the Laplace and Gaussian densities. The distance does indeed look a bit different for the Gaussian case, but this is due to a reparameterization as $\sigma$ here plays the role of $2\sigma^2$ in the standard definition of the Gaussian.

# F    MULTIDIMENSIONAL MECHANISMS

## F.1    LAPLACE MECHANISM

Suppose the summary of interest is a vector, $\mathbf{h}(D) \in \mathbb{R}^d$ with coordinates $\mathbf{h}(D) = (h_1(D), \ldots, h_d(D))$. The sensitivity of $\mathbf{h}(D)$ is $\Delta = \sup_{D \sim D'} d_{\mathbb{L}^2}(\mathbf{h}(D), \mathbf{h}(D')) = \sup_{D \sim D'} \|\mathbf{h}(D) - \mathbf{h}(D')\|_2$.

**Definition F.1.** The Laplace mechanism to release a private version of summary $\mathbf{h}(D)$ takes the form
$$\mathbf{f}(\mathbf{x}; \mathbf{h}(D), \sigma) = (f(x; h_1(D), \sigma), \ldots, f(x; h_d(D), \sigma))$$
where each $f(x; h_i(D), \sigma) = (2\sigma)^{-1} \exp\{|x - h_i(D)|/\sigma\}$ is the univariate Laplace mechanism.

That is, the Laplace mechanism adds Laplace noise to each coordinate. Since we have independence in the coordinates, computing the Rao distance is straightforward.

**Theorem F.2.** *The Laplace mechanism as in F.1 satisfies $\theta$-Rao DP with $\sigma \geq \Delta/\theta$.*

*Proof.* The proof follows directly from the formulation of product manifolds. So,

$$d(f(x; \mathbf{h}(D), \sigma), f(x; \mathbf{h}(D'), \sigma) = \sqrt{\sum_i d^2(f(x; h_i(D), \sigma), f(x; h_i(D'), \sigma))}$$

$$= \sqrt{\sum_i |h_i(D) - h_i(D')|^2/\sigma^2}$$

$$= \|h(D) - h(D')\|_2/\sigma \leq \Delta/\sigma$$

We need to bound this above by $\theta$, so we have that $\theta \geq \Delta/\sigma \Rightarrow \sigma \geq \Delta/\theta$. This completes the proof. $\square$

### F.2   GAUSSIAN MECHANISM

Suppose the summary of interest is a vector, $\mathbf{h}(D) \in \mathbb{R}^d$ with $\mathbf{h}(D) = (h_1(D), \ldots, h_d(D))$. The sensitivity of $\mathbf{h}(D)$ is $\Delta = \sup_{D \sim D'} d_{\mathbb{L}^2}(\mathbf{h}(D), \mathbf{h}(D')) = \sup_{D \sim D'} \|\mathbf{h}(D) - \mathbf{h}(D')\|_2$.

**Definition F.3.** The Gaussian mechanism to release a private version of summary $\mathbf{h}(D)$ takes the form

$$\mathbf{f}(\mathbf{x}; \mathbf{h}(D), \sigma) = (f(x; h_1(D), \sigma), \ldots, f(x; h_d(D), \sigma))$$

where each $f(x; h_i(D), \sigma) = (2\pi\sigma^2)^{-1/2} \exp\{(x - h_i(D))^2/2\sigma^2\}$ is the univariate Gaussian mechanism.

The Gauss mechanism adds Gaussian noise to each coordinate. Again leaning on independence one can utilize the product manifold formulation to determine the Rao distance.

**Theorem F.4.** *The Gaussian mechanism as in F.3 satisfies $\theta$-Rao DP with $\sigma \geq \Delta/\theta$.*

*Proof.* The proof is nearly identical to F.2 We have,

$$d(f(x; \mathbf{h}(D), \sigma), f(x; \mathbf{h}(D'), \sigma) = \sqrt{\sum_i d^2(f(x; h_i(D), \sigma), f(x; h_i(D'), \sigma))}$$

$$= \sqrt{\sum_i |h_i(D) - h_i(D')|^2/\sigma^2}$$

$$= \|h(D) - h(D')\|_2/\sigma \leq \Delta/\sigma$$

We need to bound this above by $\theta$, so we have that $\theta \geq \Delta/\sigma \Rightarrow \sigma \geq \Delta/\theta$. This completes the proof. $\square$

