# OpenReview forum: "Rao Differential Privacy"
_ICLR.cc/2026/Conference — Submitted to ICLR 2026_

### Official Review · Reviewer_tC4K · 2025-10-23

**Soundness:** 3
**Presentation:** 3
**Contribution:** 2
**Rating:** 4
**Confidence:** 3

**Summary:**

This paper introduces a new variant of Differential Privacy (DP) called “Rao Differential Privacy” (Rao DP). Instead of defining DP via a divergence of probability densities (as most other definitions do, with the exception of GDP), Rao DP defines privacy via the Rao distance of probability densities. The authors demonstrate that Rao DP supports tight (sequential) composition, and give Rao DP guarantees for the Laplace, Gaussian and Generalized Gaussian mechanisms.

**Strengths:**

1. Defining DP via distance between densities rather than a divergence makes sense.
2. The paper is well-written in a pleasant style that makes it easy to follow.
3. I think it can be argued that the paper fits the scope of ICLR.

**Weaknesses:**

1. A paper whose main contribution is to propose a new DP variant needs to, in my opinion, argue convincingly for why the new definition is needed. Besides being of theoretical interest, I do not see what need Rao DP addresses.
2. The existence of Gaussian DP weakens the case for Rao DP. While the definitions are different, one capturing distance over densities and the other hardness of distinguishing two densities via hypothesis testing, they appear to behave similarly. Both definitions exactly characterize the Gaussian mechanism (it is $\mu$-GDP and $\theta$-Rao DP), exhibit the same composition behavior, and both avoid the use of divergences in their definition.
3. I am not convinced that Rao DP as a metric for privacy is easy to convey. Due to the interpretation via hypothesis testing, 1-GDP has a clear meaning independently of the specifics of the use-case. It is not clear to me what guarantee I am afforded by 1-Rao DP.

**Questions:**

The paper is well-written and pleasant to read, and its proposed privacy definition of Rao DP appears technically correct. That said, I am not fully convinced that Rao DP as a concept has a clear use case, therefore I lean slightly towards rejection as things stand. I am willing to change my stance if the authors or other reviewers bring up good arguments in support of the definition's utility.

I ask the following questions to get a better idea of the paper’s contribution.
1. What is the use case for Rao DP? Is it a promise of tighter composition in general? If so, the paper would benefit from an example where it improves over e.g., GDP.
2. There are no conversion theorems in the paper from Rao DP to other DP variants (or vice versa). Is this endemic to how Rao DP is defined (the avoidance of divergence), or is it possible to give meaningful conversion theorems? In the case of Gaussian DP there are conversion results from Gaussian DP to zCDP and approximate DP.
3. You analyzed the Laplace and Gaussian mechanism, for which you could derive the Rao DP guarantee cleanly. Do you expect the guarantee for other DP primitives such as randomized response, sparse vector technique, report noisy max, exponential mechanism etc., to follow as cleanly? Generally speaking, is proving Rao DP for an algorithm easier or harder compared to e.g., approximate DP?
4. Conversely, is there any structure in the definition of Rao DP that make it amenable to showing strong lower bounds?
5. How would you explain the guarantee afforded by 1-Rao DP, without discussing the output distribution of the algorithm satisfying it?
6. On page 3, you have a footnote saying you only consider sequential composition, and not e.g., parallel composition. Does Rao DP satisfy parallel composition? I would assume it does, and if so I would add a proof/remark somewhere in the paper, or just add to the footnote "[...] which Rao DP also supports".

---

> ### Author Response · Authors · 2025-11-15
>
> Firstly, we thank the reviewer for their thoughtful and thorough summary and review.
>
> ``A paper whose main contribution is to propose a new DP variant needs to, in my opinion, argue convincingly for why the new definition is needed. " This is a great point, and perhaps we did not add enough justification. I disagree that $\mu$-GDP, in a sense, weakens Rao DP, but rather these can complement each other very well. For instance, for a generic mechanism to satisfy $\mu$-GDP one must carefully consider the trade-off function which can be generally difficult and then compare said function to the Gaussian trade-off. One can alternatively consider computing the Rao distance. Admittedly, either of these endeavors can be challenging but this is exactly why alternate formulations are useful. Further, we note that the Wasserstein, Total Variation, Kullback-Leibler, f-divergences, and R\'enyi divergence have been considered; there is a gap which should be examined and that is the Rao distance.
>
> Tight composition is indeed a case for Rao-DP. It does not, however, improve over GDP but rather matches it exactly. This thus gives an alternative approach to the hypothesis-testing interpretation. Our definition combines the interpretability of original DP definitions (the mechanism is not to different under adjacent datasets) with the modern composition results. We see this a huge achievement as it unifies two favorable properties. The reviewer mentioned we have no conversion theorems but we note that corollary 4.5 and 4.9 do this for specific mechanisms. For more general results we have rely on the identities $$D_{KL}(f\|g)\approx \frac{1}{2}d_{Rao}^2(f,g)$$ and $$D_{Renyi,\alpha}(f\|g)\rightarrow D_{KL}(f\|g)$$ as $\alpha\rightarrow 1$. We can make these expression more clear.
>
> Yes, Laplace, Gaussian, and Generalized Gaussian do have clean formulations for Rao DP. We did consider more general cases such as exponential mechanism and can include such results. We hadn't considered the other cases but randomized response seems clear as the Rao distance for discrete distributions are well studied. In the end, we decided on the three included mechanisms as they likely the most prevalent but concede more is always better.
>
> ``Generally speaking, is proving Rao DP ... easier or harder...?" Unfortunately the answer is that it depends. For parametric families there is literature which explores these cases and thus no work would need to be done. Since the Rao distance is a huge interest in information geometry there is a vast amount of literature which one can lean on. Yes, we can also use parallel composition but we wanted to be clear on the type of composition results which we presented. We can add the suggested theorem as it is quite simple.
>
>
> Can the reviewer elaborate on their question of lower bounds?

---

> > ### Comment · Reviewer_tC4K · 2025-11-19
> >
> > I thank the authors for their prompt and detailed rebuttal, and will take their response into consideration.
> >
> > On the point of conversion results, I do not necessarily find the absence of conversion results an issue, if there is a fundamental reason for their absence. I agree that the relation between KL divergence and Rao distance can be useful in deriving some form of conversion, and that Corollary 4.5 and 4.9 can be interpreted as conversion results in a limited sense.
> >
> > On the point of lower bounds, my question was essentially if Rao DP could be a more useful framework for deriving lower bounds for specific problems. E.g., "Any $\theta$-Rao DP algorithm for problem X incurs at least error $\Omega(\dots)$". I have no intuition for whether Rao DP as a framework is more amenable to proving lower bounds, and I would not be surprised if techniques like fingerprinting codes and packing arguments [1] can more or less directly be translated over. My point is only that, even if Rao DP offers limited benefits over GDP for tightly analyzing the privacy guarantees of mechanisms, perhaps working directly with the Rao distance offers new venues for proving lower bounds for Rao DP mechanisms. If so, that could be a potential merit of Rao DP as a framework.
> >
> > [1] The Complexity of Differential Privacy, Salil Vadhan, 2017.

---

### Official Review · Reviewer_Kvp6 · 2025-10-29

**Soundness:** 3
**Presentation:** 2
**Contribution:** 3
**Rating:** 4
**Confidence:** 3

**Summary:**

The paper proposes Rao Differential Privacy (Rao-DP), which bounds the Fisher–Rao (Rao) geodesic distance between output distributions of a mechanism on adjacent datasets using a privacy budget θ. It establishes sequential composition, post-processing invariance, and calibration for Laplace, Gaussian, and Generalized Gaussian mechanisms. The authors also relate Rao-DP to Gaussian DP (GDP).

**Strengths:**

S1: This paper introduces a true distance metric (Fisher–Rao) for privacy, linking DP to information geometry.

S2: The paper proves post-processing invariance using the square-root density embedding.

S4: It derives calibration formulas for standard mechanisms (Laplace, Gaussian).

S5: The paper extends to generalized Gaussian mechanisms beyond standard distributions.

**Weaknesses:**

W1: The definition of sensitivity in the paper is presented abstractly through a general distance function $d_\eta$ between datasets, but there is limited discussion on how this metric should be chosen or computed for real-world queries. In standard differential privacy, sensitivity is explicitly tied to the query function (for example, using L1 or L2 norms for numerical queries), which enables straightforward calibration of noise. While the Rao-DP formulation leaves $d_\eta$ largely symbolic. It would be more interesting to make it clear how to apply the framework to practical tasks such as mean estimation, histogram queries, or model parameter updates.

W2: The current explanation of $\delta$ as “the probability that pure-DP holds with probability 1 − $\delta$ and is violated with probability $\delta$” is somewhat misleading. $\delta$ does not literally represent the probability that privacy fails on a random draw. Rather, $\delta$ bounds the total probability mass of outcomes for which the $\epsilon$-DP inequality may not hold. It would be clearer to phrase $\delta$ as a worst-case slack or tail-probability bound instead of a “failure probability.”

W3: There are many typos and grammar errors:
Definition 2.1: “is said to by $\epsilon$-differentially private” → should be “is said to be $\epsilon$-differentially private.”
Definition 2.2: “datesets” → should be “datasets.”
“a pre-specified parameters” → should be “pre-specified parameters.” (remove “a”).

Page 3: "One requires the defintion of" -> "One requires the definition of"
Page 9: "definition an be extended" -> "definition can be extended"

**Questions:**

Q1: Could the authors clarify how the distance function $d_\eta$ is chosen or computed in practice? For instance, should it measure differences in the data domain, the mechanism’s output space, or the Fisher–Rao parameter space?

Q2: Could the authors clarify their interpretation of the $\delta$ parameter in Definition 2.2? Specifically, do they intend $\delta$ to represent a literal probability of privacy failure, or a bound on the probability mass where the $\epsilon$-DP inequality may be violated?

Q3: Does the √(Σ θᵢ²) composition bound rely on independence between mechanisms? If the mechanisms are dependent or adaptive, does the bound still hold, and under what conditions?

---

> ### Author Response · Authors · 2025-11-15
>
> We would like to thank the reviewer for their thorough evaluation. We give response to the weaknesses and questions below.
>
> W1: We understand that the sensitivity is a bit abstract. As the reviewer rightly pointed out, sometimes sensitivity requires L1 or L2 norms (distances) for Laplace and Gaussian, respectively for instance. Our intention was simply to compactify the different definitions into one. The norm is, in a sense, determined by the choice of mechanism so we kept it as an abstract distance to not require separate definitions of sensitivity. Further, we note that this notation is not entirely uncommon. In [1] (see definition 2), they require the L$\beta$ sensitivity.  Our notion of sensitivity generalizes this even further to accommodate the sensitivity as in [2].
>
> W2,Q2: Thank you for pointing out this flaw, another reviewer recognized this as a flaw as well. We note though that the language we used is similar to that of [3], a paper we heavily cited. We agree with the reviewers, though, that our particular phrasing can be misleading.  We will work to make this statement more clear.
>
> W3: We apologize that the grammar and spelling leave much to be desired. Thank you for pointing out the specific errors as this information is valuable for a resubmission. We will heavily review and edit the grammar.
>
> Q1: We touched on this a bit in W1 but we elaborate more here. In the vast landscape of DP, there are many definitions of sensitivity which arise. For example, $\|h(D)-h(D')\|\_1$, $\|h(D)-h(D')\|\_2$, $\| E(\theta;D)-E(\theta;D') \| $ (as in the exponential mechanism), $\| \nabla E(\theta;D)-\nabla E(\theta;D') \|\_K$ (as in the K-norm gradient mechanism [4]), and $d_{\mathcal{M}}(h(D),h(D'))$ (as in the manifold Laplace);
>
>  while these are all fairly different, they share the idea of measuring \textit{some} distance between summaries (or energies). Our $d_\eta$ was meant to capture the idea that sensitivity does not have a ``one-size-fits-all" norm, but rather it is dependent on the mechanism. As in W1, we again mention that our definition of sensitivity matches recent DP papers such as [1], but we are even more general to accommodate DP defined on non-linear spaces such as the manifold Laplace. Lastly, in [5] Dwork had two definitions of sensitivity as well, first the L1 sensitivity and a general metric space sensitivity.
>
>
>
> Q3: Great question. The proof does rely on independence, as this is how we invoke the product manifold characterization. We hadn't considered the dependent case, but in this case we are presenting an upper bound. We will add details on this.
>
>
> 1. Roy Rinberg, Ilia Shumailov, Vikrant Singhal, Rachel Cummings, and Nicolas Papernot. Beyond laplace and gaussian: Exploring the generalized gaussian mechanism for private machine learning. arXiv preprint arXiv:2506.12553, 2025
> 2. Matthew Reimherr, Karthik Bharath, and Carlos Soto. Differential privacy over riemannian manifolds. Advances in Neural Information Processing Systems, 34:12292–12303, 2021
> 3. Ilya Mironov. R´enyi differential privacy. In 2017 IEEE 30th computer security foundations symposium (CSF), pages 263–275. IEEE, 2017
> 4. Matthew Reimherr and Jordan Awan. Kng: The k-norm gradient mechanism. Advances in Neural Information Processing Systems, 32, 2019
> 5. Cynthia Dwork, Frank McSherry, Kobbi Nissim, and Adam Smith. Calibrating noise to sensitivity in private data analysis. In Theory of cryptography conference, pages 265–284. Springer, 2006

---

> > ### Comment · Reviewer_Kvp6 · 2025-11-26
> > **Response to the authors**
> >
> > Thank you for the detailed rebuttal. I will take the authors’ responses into consideration.
> >
> > I appreciate the clarifications on (i) the intention behind keeping sensitivity abstract (W1/Q1), (ii) the plan to revise the phrasing around $\delta$ to avoid “failure probability” confusion (W2/Q2), and (iii) the commitment to a careful grammar and typo pass (W3). That said, I believe the paper would be strengthened by including at least one fully worked-out instantiation of the abstract sensitivity $d_\eta$ for a standard mechanism.

---

> > > ### Author Response · Authors · 2025-11-26
> > >
> > > Thank you for the reply.
> > >
> > > For the Laplace we do implicitly work out the sensitivity in Theorem 4.4, but we will be clearer. We have that $\Delta=\max_{D\sim D'} |h(D)-h(D')|$ for the Laplace where the max (or sup as necessary) is used as we need the bound to hold for all adjacent datasets. We do see that the connection to the original definition of sensitivity is not straightforward. Here we have $\Delta=\max_{D\sim D'} |h(D)-h(D')|=\max_{D\sim D'} d(h(D),h(D'))$ where the distance is L1 (which coincides with L2 since these are real value not vector valued).
> > >
> > > We will make this clearer and add the Gaussian and generalized Gaussian sensitivity.

---

### Official Review · Reviewer_BYT6 · 2025-10-30

**Soundness:** 3
**Presentation:** 3
**Contribution:** 3
**Rating:** 6
**Confidence:** 3

**Summary:**

This paper introduces Rao Differential Privacy (Rao DP), a new definition of differential privacy based on the Rao distance rather than divergence measures such as KL or Rényi divergence. The authors take an information geometry perspective, proposing that measuring the dissimilarity between mechanisms via a proper distance metric offers a more natural and interpretable notion of privacy. They show that Rao DP satisfies key properties of DP, i.e., post-processing immunity and sequential composition, and derive the corresponding privacy parameters for Laplace, Gaussian, and Generalized Gaussian mechanisms. The composition rule under Rao DP yields a tighter bound (Euclidean-like addition of privacy budgets) than traditional DP.

**Strengths:**

* Introduces a novel, geometry-based definition of differential privacy using the Rao distance.

* Demonstrates that Rao DP satisfies composition and post-processing properties fundamental to DP.

* Provides closed-form derivations for Laplace, Gaussian, and Generalized Gaussian mechanisms.

**Weaknesses:**

* Limited empirical or practical validation

The paper is entirely theoretical; it does not provide any numerical illustration or simulation to demonstrate the implications of Rao DP in realistic DP tasks (e.g., trade-offs between privacy and utility). Including such an example would make the contribution more compelling.

* Comparative discussion lacks depth

While the paper claims tighter composition and better interpretability, it does not quantitatively compare the bounds of Rao DP with those of existing relaxations (e.g., Rényi DP or zCDP) on a common task. Without this, it is hard to assess how substantial the improvement is in practice.

* Connection to Gaussian DP is underdeveloped

The paper notes that Rao DP and Gaussian DP (GDP) share similar composition behavior and even the same parameter mapping for the Gaussian mechanism. However, the conceptual difference between the two frameworks remains somewhat vague. The discussion in Appendix C.1 hints at a deeper connection but stops short of establishing whether Rao DP subsumes or generalizes GDP.

* Scope limitations

The definition is demonstrated only for continuous, one-parameter mechanisms. The extension to multivariate or discrete domains is acknowledged but not developed, which limits the generality of the proposal.

**Questions:**

1. The paper claims tighter composition, but could the authors provide a quantitative example comparing the total privacy loss between Rao DP and Rényi DP or zCDP for the same mechanism and parameters?

2. How does Rao DP relate to hypothesis-testing interpretations of privacy? Can the geometric interpretation yield new privacy–utility trade-offs beyond composition?

3. Since GDP and Rao DP coincide for the Gaussian mechanism, is Rao DP effectively a geometric reinterpretation of GDP or does it generalize it to other mechanisms?

---

> ### Author Response · Authors · 2025-11-15
>
> Firstly, we thank the reviewer for their thoughtful and thorough summary and review.
>
> While it is true that our paper is entirely theoretical, the composition result has some clear empirical implications. Suppose one were to use the Laplace to release 10 estimates each with 0.1 budget, under pure DP the total budget is 1 but under Rao DP the total budget is $\sqrt{10*.1^2}\approx  0.3162278$. This is similar to the contribution of $\mu$-GDP [1], as they also give a different perspective while allowing the use of familiar mechanisms.
>
>
>
> The relationship between hypothesis testing interpretation and Rao DP is quite interesting. Due to space limitation we put some discussion of this in Appendix C near the end as the reviewer pointed out. In short, in [2] Rao proposes a test statistic for a hypothesis test which utilizes the Rao distance as the numerator. Informally, in the [1] section ``Distance in Tests of Significance and Classification" Rao proposes that $t_s=\frac{\hat{d}_R(f,g)}{Var(\hat{d}_R(f,g))^{1/2}}$ where I am using the hat to represent the distance of an observed sample. Now this leaves the question what is the hypothesis-testing DP interpretation test statistic? This is unclear to us. It seems that $\mu$-GDP relies on Neyman-Pearson and states that a test exists but not what it is.
>
> We are unclear on the extension to multivariate as we have achieved that via the product manifold. We can make this more clear as we agree higher dimension mechanisms are clearly of importance. For discrete domains it is a bit more challenging but closed-form expressions for the Rao distance exist. While I agree this can add substance to this paper, we believe that the Laplace, Gauss, and Generalized Gauss are crucial cornerstones for DP and this paper can motivate future extensions.
>
> 1. Jinshuo Dong, Aaron Roth, and Weijie J Su. Gaussian differential privacy. Journal of the Royal Statistical Society: Series B (Statistical Methodology), 84(1):3–37, 2022
> 2. C Radhakrishna Rao et al. Information and the accuracy attainable in the estimation of statistical parameters. Bull. Calcutta Math. Soc, 37(3):81–91, 1945

---

### Official Review · Reviewer_zqZG · 2025-10-30

**Soundness:** 3
**Presentation:** 2
**Contribution:** 2
**Rating:** 2
**Confidence:** 4

**Summary:**

The paper introduces a new notion of differential privacy namely Rao DP

**Strengths:**

See detailed comments below

**Weaknesses:**

See detailed comments below

**Questions:**

This paper introduces a new definition of privacy based on the Rao distance for densities proposed in [1].

Major comments

1) The Gaussian and Laplace mechanisms are computed for the 1 dimensional manifold of distribution (with parameter fixed) with results from [1]. However it is not properly cited i.e., it is not apparent that the equations given in Sec 2.3 Example 1 have been derived in [1] when it is the case. Please refer to them like "from the result in Section 4.2 of [1]". The same is the case in Sec 3.1 (the equation just above Lemma 3.2) where a result from [1] is directly written as well as applied in Lemma 3.2. Wherever direct equations from a source paper are re-written, it is expected to cite the source clearly.

2) Apart from the post-processing proof Sec 3.2, the paper lacks novelty of theoretical contributions, although the idea of using the Rao distance for DP is interesting. The theoretical analyses could have been extended for a 2-dimensional manifold, as most of the results for 1-dimensional case has been derived in previous works already as pointed out above

3) Line 453 seems to be a misleading statement and seems to indicate a strong advantage for the Rao DP framework which is incorrect. Gaussian noise mechanisms will lead to non-zero privacy leakage since the privacy loss RV is unbounded. By changing  the notion of DP to the Rao DP framework one cannot  claim the following statement "As opposed to approximate DP, we do not have a second
parameter such as $\delta$. This is a strength of our framework since, as noted previously, this $\delta$ is the
probability of privacy leakage. Our formulation does not allow such a case" (line 453-455 of the paper). In fact most works in DP usually give a conversion from their notion of DP to the standard $(\epsilon, \delta)$ notion of DP since that is the widely accepted operational notion of DP. Gaussian mechanism will lead to non-zero privacy leakage and the fact that Rao DP does not reveal this is actually worrisome.

4) The impact of the derived results is not clear since  here differential privacy (ie Rao DP) is defined in
terms of a distance metric for densities while other notions are divergence based, hence, how will one compare across these different notions of DP?  To get a better perspective some empirical results would help. It would be useful to see an application where this notion of Rao DP helps


minor comments
1) At certain places, the writing style of the paper is not formal. For example, in the paragraph just below Definition 2.3, the sentence "More on this shortly." is written, not referring to the exact section or subsection to point the reader towards.
2) I found typos/grammatical errors in few following places: 4th line, 2nd paragraph of Sec 1; 7th line, 3rd paragraph of Sec 1; 1st line of Definition 2.1 in Sec 2.1Please do a proof check for grammatical errors.

3) In 3rd paragraph of Sec 2.3, shouldn't the "not belongs to" symbol be used for p_1 + p_2 ?

[1] Burbea, J., Rao, C.R.: Entropy differential metric, unified approach. Journal of Multivariate Analysis 12, 575–596 (1982)

---

> ### Author Response · Authors · 2025-11-15
>
> Firstly, we thank the reviewer for their thoughtful and thorough review.
>
> We are a bit unclear about the improper citation statement but would gladly add said citation. The reviewer has pointed at a citation (including Rao) from 1982, but we cited Rao's 1945 paper. Our example from sec 2.3 is indeed included in the 1945 paper. As for the equations from Sec 3.1, this result is a classical result in Riemannian geometry, we touch on this, and refer to this, in Appendix A. We thank the reviewer for pointing us toward the citation and are happy to include it but we note that we do cite [1], [2], and [3] as texts where these equations also exist.
>
> Thank you for noting that the Rao distance for DP is an interesting idea. In terms of higher dimensional manifold we must be careful so as to still be aligned with previous notions of privacy. If by higher dimensions we mean privatizing $h(D)=[h_1(D),h_2(D)]$, we did consider this case by our composition result which is for product manifolds and hence we can handle $m$-dimensional. To satisfy the previous definitions of privacy $\sigma$ must be some minimal size; for higher dimensional manifolds on could consider the matrix $\Sigma$ but then there would be many questions as to how to define a minimal size, perhaps determinant or Frobenius norm, but even then this would imply that the mechanisms are not necessarily independent which could be problematic. For diagonal $\Sigma$ again this is just the case of a product manifold.
>
>
> We apologize that line 453 is misleading. We do note that our
> Corallary 4.9 actually does allow us to to do a conversion. Just as in the conversion of $\mu$-GDP  to approx. DP one would need to pick and $\epsilon$ or $\delta$ to get the final parameter. We should be more clear, similar to how $\mu-$ GDP has one parameter we also have one parameter. This leads to more than one possible approx. DP setting but note that each approx DP setting is exactly one Rao DP.
>
> To compare across different notions of DP one has a few options such as (1) they can determine the Rao distance for the specific mechanism and compare to the results from their notion of DP (i.e. compare what the spread parameter must be), or (2) they can rely on identities such as $$D_{KL}(f\|g)\approx \frac{1}{2}d_{Rao}^2(f,g)$$ and $$D_{Renyi,\alpha}(f\|g)\rightarrow D_{KL}(f\|g)$$ as $\alpha\rightarrow 1$. There are several options.
>
>
> We thank the reviewer for the minor comments. Yes, indeed it should be $p_1 + p_2 \notin \mathcal{P}$. We will edit the paper to fix the grammar and typos.
>
> 1. C Radhakrishna Rao et al. Information and the accuracy attainable in the estimation of statistical parameters. Bull. Calcutta Math. Soc, 37(3):81–91, 1945
> 2. Manfredo Perdigao Do Carmo. Riemannian geometry, volume 2. Springer, 1992
> 3. John M Lee. Introduction to Riemannian manifolds, volume 2. Springer, 2018

---

> > ### Comment · Reviewer_zqZG · 2025-11-27
> >
> > I thank the authors for their response
> > However, I reiterate that the proofs and technical contributions are quite limited, as the core arguments have already been established in the one-dimensional Gaussian setting. It is not just that the work [1] was missing in the list of references; many results stated in the manuscript directly follow from [1]. As a result, the overall novelty of the work is limited.
> >
> > Further I also reiterate part of point 4 I had raised before namely, yt would be useful to see an application where this notion of Rao DP helps
> >
> > [1] Burbea, J., Rao, C.R.: Entropy differential metric, unified approach. Journal of Multivariate Analysis 12, 575–596 (1982)

---

### Official Review · Reviewer_duK9 · 2025-11-04

**Soundness:** 2
**Presentation:** 3
**Contribution:** 2
**Rating:** 4
**Confidence:** 3

**Summary:**

The paper introduces a new definition of differential privacy (DP) from an information geometry perspective. All existing definitions of differential privacy (like standard $\epsilon$-DP, Rényi DP, etc.) are based on measuring the "difference" between a mechanism's output on adjacent datasets using a divergence (like the Kullback-Leibler (KL) divergence), which are often asymmetric. The paper proposes a new definition that measures this difference using a proper distance metric (which is symmetric). Specifically, it uses the Rao distance. The core contribution is to introduce Rao DP - a novel DP definition that uses the Rao distance as its measure of similarity. It proves RDP has crucial DP properties: post-processing privacy preservation and sequential composition of privacy budgets. Finally, it analyzes classic mechanisms: The paper determines the privacy parameters (the "privacy cost") for the two most common mechanisms, the Gaussian mechanism and the Laplace mechanism, under this new RDP framework.

**Strengths:**

This paper has conceptual novelty. To some extent it challenges the existing

**Weaknesses:**

Despite the novelty I'm not fully convinced that we should bring such a new definition into the already rich collection of potentially meaningful DP definitions. The paper correctly points out that the previous definitions do not really use metrics, but I don't think that is crucial if we aren't looking for purely rigorous mathematic definitions. All these definitions are still "symmetric" in the sense that the same constraints imposed by privacy need to hold even if we swap $D$ and $D'$. It's also unclear how the Rao metric compares to other metrics. Does it guarantee closeness of the distributions everywhere? Is it more (or less) strict when compared to previous definitions? The paper gave some discussion but it lacks depth. Finally there is also the question of what we can gain from this new proposal. The paper used it to sort of paraphrase the privacy guarantees of established privacy mechanisms. The results are not surprising. It's unclear whether it can be used to solve real problems.

**Questions:**

See Weakness. In addition:
1. Have you considered using other distance metrics between distributions as well? Rao metric is not the only one that makes sense from that perspective.
2. Does the new definition bring about new mechanisms designs, understanding of previously existing problems, etc.?

---

> ### Author Response · Authors · 2025-11-15
>
> Firstly, we thank the reviewer for their thoughtful and thorough summary and review.
>
> In terms of the summary we point out that besides the Laplace and Gaussian mechanism, we also determine the privacy parameters for the Generalized Gaussian. This section, however, is in the appendix due to space limitations. While the Generalized Gaussian is a lesser known mechanism, it has gained some attention recently [1] as it is quite versatile and flexible.
>
> Weaknesses:
>
> We agree that the currently collection of DP definitions is currently quite rich, but from our perspective it is incomplete. We also agree that by taking the supremum over adjacent datasets, the previous definitions are symmetrised. In this paper we explored the questions ``What happens when we use a metric and thus remove the need to symmetrize?" To answer this question we surveyed the landscape of available metrics for distributions/densities and found that the Rao distance is quite special. The Rao distance is invariant to reparametrization and invariant to sufficient statistics; these are crucial property when we consider information loss but unfortunately we do not emphasize in the paper. We can add a longer discussion on this upon resubmission.
>
> As for, ``how the Rao metric compares to other metrics," there are many interesting identities which we can include in the appendix. For instance, $$D_{KL}(f\|g)\approx \frac{1}{2}d_{Rao}^2(f,g)$$ and $$D_{Renyi,\alpha}(f\|g)\rightarrow D_{KL}(f\|g)$$ as $\alpha\rightarrow 1$; together these two identities tie together many of the definitions. These allow us to directly compare the Rao distance to the KL and R\'enyi divergences and hence the classical DP defintions and lean on their interpretations.
>
> As we see it, towards the question of  what we can gain,' we see our definition as a way to unify (1) the interpretability of classical DP definitions i.e. the mechanism is not too different under adjacent datasets, and (2) the tighter composition results of new DP definitions i.e. the composition we see in, for instance, $\mu-$GDP. To our knowledge, this is the only way to achieve the tightest composition while ``avoiding" the hypothesis-testing interpretation. Further, while the $\mu-$GDP formulation is a huge achievement, it requires (1) working with a trade-off function of the mechanism, and (2) comparing to the Gaussian trade-off function. Completing these tasks for general mechanisms is not a simple undertaking, so our definition allows an alternative solution.
>
>
> As we alluded to earlier, yes, we did consider other distances. The total variation has been considered in [3] and Wasserstein in [4]. The Mahalanobis distance is an important distance and, in fact, the Rao distance is a generalization of the Mahalanobis; these two definitions coincide in certain scenarios. The Hellinger distance would be an interesting choice which we considered as well. The Hellinger distance and Rao distance are also very well connected. Note the Hellinger distance square-roots the densities, effectively embedding the density onto a sphere. The Hellinger distance is thus the chordal distance while the Rao distance is the arc length distance. All this to say, we are filling a gap in the literature and we see this a necessary contribution for a complete DP landscape.
>
> 1 Roy Rinberg, Ilia Shumailov, Vikrant Singhal, Rachel Cummings, and Nicolas Papernot. Beyond laplace and gaussian: Exploring the generalized gaussian mechanism for private machine learning. arXiv preprint arXiv:2506.12553, 2025
>
> 2 Nihat Ay, J\¨urgen Jost, H\ˆong V\ˆan L\ˆe, and Lorenz Schwachh¨ofer. Information geometry and sufficient statistics. Probability Theory and Related Fields, 162(1):327–364, 2015
>
> 3  Rina Foygel Barber and John C Duchi. Privacy and statistical risk: Formalisms and minimax bounds. arXiv preprint arXiv:1412.4451, 2014
>
> 4 Aman Bansal, Rahul Chunduru, Deepesh Data, and Manoj Prabhakaran. Extending the foundations of differential privacy: Flexibility and robustness. 2020

---

### Author Response · Authors · 2025-12-03

Area Chairs and Senior Chairs,

Given the privacy leak and freezing of discussion we give a brief summary of the discussion below. First, we mention commonly stated weakness of the paper.

* **The need for a metric based definition.** This was a common issue brought up by the reviewers. From our standpoint we are accomplishing 2 things, first we are filling a gap in the literature and second, we are unifying two frameworks. In terms of filling a gap we mean that DP has considering many ways to measure similarity of densities to define DP including the most common such as Kullback-Leibler divergence, Total Variation, and Renyi divergence. However, in terms of measuring information it is missing, for instance, the Rao distance. This paper begins to answer the questions of how the privacy implications change under this setting. In terms of latter we mean that original pure DP effectively states *the densities are not too different under adjacent datasets,* this interpretation is very intuitive. There have been years of research on how to ``improve" DP namely on how can one get better composition results. A main contender for composition at the moment are Gaussian DP; Gaussian DP and Rao DP match in composition results but Rao DP has the added benefit of retaining the original DP notion of similarity of densities. We see these two accomplishments as proper justification for pursuing this route.
* **The description of delta.** Our original description of delta was misleading. we have since fixed it. We had used informal language similar to other DP papers to describe delta but have since rewritten this description to better describe delta.

Following we summarize the discussion between each reviewer.

---

> ### Author Response · Authors · 2025-12-03
>
> * **Reviewer duK9:** The reviewer mentioned a few weaknesses 1. the reason for introducing a metric based definition (including what is gained from this definition)  and 2. the comparison of the Rao metric to other metrics. For 1. we believe the above answers part of their question. The reviewer also wishes to know if this leads to new insight on mechanism design or novel ways to tackle existing problems. In short we say yes but to tackle all such things in one paper would be a large undertaking. We show three mechanisms in our paper Laplace, Gaussian, and generalized Gaussian, and we believe this is a great starting point for further exploration. For 2. We included some conversions in the rebuttal which we will include upon resubmission.
>     **The reviewer gave us an initial score of 4, the reviewer did not have a chance to respond to our rebuttal before the data leak incident.**
> * **Reviewer zqZG:** The reviewer had four major comments. 1. Citation issue in conjunction with novelty associated with previously derived results, 2. Novelty other than post-processing, 3. The description of delta and the fact that it is missing, and 4. The impact of Rao DP. For 1. we were slightly unclear at first. We noted that many of the results are fairly standard in differential geometry, but we agree that upon re-reading some things were left unattributed. We have since edited the paper to add more citations. For 2. we were unsure which way the reviewer envisioned a higher dimensional manifold. We have since added appendix section F which handles vector valued statistics. We didn't originally have this as it follows directly from other parts of the paper but now it is more clear. For 3. and 4. we address as above. 3. also mentioned that the lack of delta is concerning but we noted that GDP also does not have a delta. We, just like GDP, include a conversion at least for the Gaussian case. The reviewer also had some minor comments which we have partially addressed in the new version of the paper. We are still editing the paper however.
>     **The reviewer gave us an initial score of 2. The reviewer did not believe our work was novel enough and did not adjust their score. We however feel we have addressed their major concerns.**
> * **Reviewer BYT6:** The reviewer represented the following weaknesses 1. Limited empirical/practical validation, 2. Comparative discussion lacks depth, 3. connection to GDP underdeveloped, and 4. Scope limitations. For 1. we do note that our paper is purely theoretical. The implications of this paper are rather that we can continue to conduct DP in the usual sense but with different budget accounting and further another tool for analysis. For 2. we did not provide a sufficient answer, and we agree this would strengthen the paper. We do see this as a potential future research avenue. For 3. We agree the connection is a bit underdeveloped but this is mainly due to the connection being unclear to begin with. GDP utilizes the hypothesis test viewpoint without proposing a test statistic, so it is unclear to us exactly how these are connected. We do however mention in our rebuttal how the Rao distance is used as a test statistic. As Rao DP utilizes a similarity of densities and GDP utilizes the hypothesis test viewpoint, we are not sure how to fully develop a connection other than what we have already shown. Lastly for 4, we have included appendix F to tackle vector valued statistics. This was not included in the earlier submission as it follows directly from the previous parts of the paper.
>     **The reviewer gave us an initial score of 6. The reviewer did not have a chance to respond to our rebuttal before the data leak incident.**
> * **Reviewer Kvp6:** The reviewer pointed at 3 major weaknesses 1. the abstract definition of sensitivity, 2. the characterization of delta, and 3. the grammar. For 1. we mention that the abstractification of sensitivity has been a recent development and pointed at recent papers with similar definitions. We agree it is not entirely clear and will include a clearer connection between our definition and the more typical use. For 2. we address as above. For 3. We have fixed many of the grammar issues on our new submission but will continue to edit it.
>    **The reviewer gave us an initial score of 4. The reviewer mentioned they would take our response into consider as it seems we have addressed their major concerns.**

---

> > ### Author Response · Authors · 2025-12-03
> >
> > * **Reviewer tC4K:** The reviewer pointed at 3 weaknesses and had a few questions as well. The weaknesses are 1. the need of a new definition, 2. GDP weakens the case for Rao DP as it matches the composition result, and 3. The privacy guarantee of Rao DP. For 1. we argue as above. For 2. this was an interesting point we had not considered. We think GDP+Rao DP work well together in the sense that in neither case may it be particularly simple to show a mechanism satisfies a definition. So, in conjunction we have two routes one could take. This gives practitioners another tool. For 3. we agree it is not a simple message to convey. For the main paper body we focus on 1-dimensional densities as it is the simplest case and even in this case it is not so straightforward. However, extensions are straightforward as we show in appendix F. The privacy guarantee does need to be expanded more but we note we have similar interpretation to epsilon indistinguishability. The reviewer also had several questions which we presumably mainly addressed.
> >     **The reviewer gave us an initial score of 4. The reviewer felt our response was detailed but also left some open questions. We do see some of the open questions and avenues for future research.**
> >
> > We hope the above summary is suitable. It is, of course, very condensed. We hope we have not misrepresented the reviewers' remarks.
> >
> > **Summary:** Initial scores were 6,4,4,4,2. The scores did not change. We have submitted an updated version of the paper including some changes the reviewers requested such as appendix F, adding references, and wording on $\delta$.

---

### Meta-Review · Area_Chair_mjE4 · 2026-01-12

**Summary:**

This paper introduces Rao Differential Privacy, a novel formulation of differential privacy based on the Rao (Fisher–Rao) distance from information geometry rather than divergence-based measures. The paper is mathematically careful and offers an elegant geometric interpretation of privacy loss as a proper metric between distributions. The authors establish core DP properties such as post-processing and sequential composition, and derive privacy parameters for common mechanisms including Laplace and Gaussian mechanisms.

Despite these strengths, I do not believe the paper meets the acceptance bar for ICLR in its current form. Across the reviews, there is broad agreement that while the proposal is conceptually interesting, it does not yet demonstrate clear operational or practical advantages over existing privacy frameworks such as Rényi DP, zCDP, or Gaussian DP. In particular, several reviewers note that for standard mechanisms, the results largely recover known guarantees under reparameterization, and that the benefits of adopting a distance-based definition remain insufficiently justified from an ML or deployment perspective. The rebuttal clarifies technical points and improves exposition, but does not fully resolve concerns about impact, scope, and relevance to typical machine-learning pipelines. I therefore recommend rejection, while encouraging the authors to further develop the operational implications and practical motivation in a future submission.

**Reviewer Concerns:**

Several concerns raised by the reviewers were addressed by the rebuttal. BYT6 asked for clarification on whether Rao Differential Privacy satisfies the core properties expected of a DP definition, which was addressed through clearer proofs of post-processing immunity and sequential composition. Kvp6 raised questions about how standard mechanisms such as Laplace and Gaussian fit within the framework, and these were addressed by explicitly deriving Rao-DP parameters and clarifying their relationship to existing DP notions. tC4K requested clearer exposition and motivation for adopting a distance-based definition, which was partially addressed through expanded discussion of the geometric interpretation.

Some concerns remain outstanding. duK9 expressed skepticism about whether bounding Rao distance provides an operationally meaningful privacy guarantee comparable to likelihood-ratio or hypothesis-testing interpretations central to classical differential privacy, and this concern was not fully resolved. zqZG was the most critical, questioning the degree of novelty and arguing that much of the analysis closely parallels existing information-geometric results or Gaussian DP; despite clarifications and added citations, concerns about limited added value remain. More broadly, across reviewers, uncertainty persists about whether Rao Differential Privacy offers concrete advantages over existing frameworks in terms of utility, deployment, or new mechanisms, which ultimately motivates the rejection.

**Reviewer Scores:**

BYT6 (score 6).
This reviewer was generally positive but cautious about overall impact. After discussion, the score would likely have remained unchanged.

Kvp6 (score 4).
This reviewer acknowledged the conceptual and mathematical aspects of the work but expressed reservations about practical benefits. The score would likely have remained unchanged.

tC4K (score 4).
This reviewer viewed the paper as technically sound but below the acceptance bar. While some points were clarified, the overall assessment would likely remain similar.

duK9 (score 4).
This reviewer expressed ongoing concerns about the operational interpretation of the proposed privacy definition. The rebuttal provided clarification, but the score would likely have remained unchanged.

zqZG (score 2).
This reviewer was the most skeptical regarding novelty and impact. While some technical issues were addressed, the overall assessment would likely not change substantially after discussion.

Overall, discussion clarified the scope and intent of the contribution but did not lead to meaningful shifts in reviewer scores.

---

### Decision · Program_Chairs · 2026-01-26

Reject